# Predicting Diagnosis of Australian Canine and Feline Urinary Bladder Disease Based on Histologic Features

**DOI:** 10.3390/vetsci7040190

**Published:** 2020-11-27

**Authors:** Emily Jones, John Alawneh, Mary Thompson, Chiara Palmieri, Karen Jackson, Rachel Allavena

**Affiliations:** 1School of Veterinary Science, The University of Queensland, Gatton, QLD 4343, Australia; j.alawneh@uq.edu.au (J.A.); c.palmieri@uq.edu.au (C.P.); karen.jackson@uq.edu.au (K.J.); r.allavena@uq.edu.au (R.A.); 2Good Clinical Practice Research Group, School of Veterinary Science, The University of Queensland, Gatton, QLD 4343, Australia; 3School of Veterinary Medicine, College of Science, Health, Engineering and Education, Murdoch University, Murdoch, WA 6150, Australia; mary.thompson@murdoch.edu.au

**Keywords:** bladder, logistic regression, veterinary histopathology, predictive model

## Abstract

Anatomic pathology is a vital component of veterinary medicine but as a primarily subjective qualitative or semiquantitative discipline, it is at risk of cognitive biases. Logistic regression is a statistical technique used to explain relationships between data categories and outcomes and is increasingly being applied in medicine for predicting disease probability based on medical and patient variables. Our aims were to evaluate histologic features of canine and feline bladder diseases and explore the utility of logistic regression modeling in identifying associations in veterinary histopathology, then formulate a predictive disease model using urinary bladder as a pilot tissue. The histologic features of 267 canine and 71 feline bladder samples were evaluated, and a logistic regression model was developed to identify associations between the bladder disease diagnosed, and both patient and histologic variables. There were 102 cases of cystitis, 84 neoplasia, 42 urolithiasis and 63 normal bladders. Logistic regression modeling identified six variables that were significantly associated with disease outcome: species, urothelial ulceration, urothelial inflammation, submucosal lymphoid aggregates, neutrophilic submucosal inflammation, and moderate submucosal hemorrhage. This study demonstrated that logistic regression modeling could provide a more objective approach to veterinary histopathology and has opened the door toward predictive disease modeling based on histologic variables.

## 1. Introduction

Anatomic pathologists are highly trained in the use of light microscopy, and increasingly digital whole-slide images, for analysis of tissue specimens at 20 to 600 times magnification to evaluate the cellular components and characteristics of a disease or physiologic process. Pathologists make a diagnosis through the recognition of one or more histologic findings, ideally combined with the clinical history of the patient [1]. However, it is well known that pathologists are influenced by cognitive bias [2,3]—a facet of the discipline that has both benefits and limitations [4]. Examples of cognitive bias include (a) confirmation bias—looking for evidence of a favored hypothesis; (b) diagnostic drift—slight variations in scoring values over the course of a study; (c) tunnel vision or anchoring—the tendency to rely too heavily on the first information presented; (d) avoidance of extreme scoring ranges; and (e) availability bias—referring to what most easily comes to mind, which means less familiar diseases are not considered [3,5].

Manual histopathology is generally qualitative or semi-quantitative as opposed to quantitative, and at times has low agreement between individual pathologists [6,7] which has prompted recent advances in computer learning and computer-based image analysis [3]. Mathematical models for pathologists’ diagnostic processes do exist. However, very few have been validated [8]. When a pathologist looks at a tissue microscopically, they incorporate the observed changes with their knowledge obtained via extensive training and experience to formulate an ordered differential list [9]. The diagnosis with the highest probability is then reported as the final diagnosis. In summary, pathologists incorporate a wide range of knowledge and experience to provide the most likely diagnosis based on the microscopic changes they observe.

Logistic regression is a statistical method used to explain the relationship between data categories and outcomes [10]. The output of logistic regression is an odds ratio which reflects the likelihood that an event/disease occurs as depending on dichotomous explanatory variables (for example, what is the odds of a patient developing diabetes if they are overweight or not overweight) [11]. In a stepwise procedure, logistic regression is used to build the best possible model that explains the relationship between the outcome/s of interest and the explanatory variables in the dataset [10,11]. To do this, the model uses logit (log-odds) transformation to predict the probability of the outcome/s occurring in every possible combination of explanatory variables, thus acts as a simulation [10,11]. Logistic regression modeling was chosen for this study because it allows prediction of outcome probability from a combination of continuous and discrete independent variables [12].

Disorders of the urinary bladder are common in dogs and cats; therefore, bladder tissue is relatively easy to acquire. Bladder diseases account for 7% of cats hospitalized in veterinary clinics in the United States [13]. An estimated 14% of North American dogs are affected by a urinary tract infection at some point during their lifetime [14] and 1.5–3% of dogs admitted into veterinary care are diagnosed with urolithiasis (bladder stones) [15]. Urinary bladder neoplasms (primarily urothelial carcinoma, UC, previously termed transitional cell carcinoma, TCC) account for 1.5–2% of all canine neoplasms, while the prevalence of bladder neoplasia in cats is much lower [16]. Urinary bladder disorders can be grouped into four broad categories—neoplastic disease, urinary tract infection, non-infectious cystitis (namely feline idiopathic cystitis) and urolithiasis [17].

Statistical models have potential to predict disease risk in both the human and veterinary medical fields by providing an objective probability of the disease occurring given the combination of variables [18] In the veterinary field, statistical models have been used in genetics [19], ultrasonography [20,21], surgery and surgical prognosis development [22,23] as well as predicting disease outbreaks [24]. However, the use of statistical models in veterinary histopathology has been limited [25]. The aims of this study were to evaluate the microscopic changes of urinary bladder diseases in a retrospective collection of canine and feline urinary bladder tissue and to quantify the association, if any, between the microscopic changes and disease diagnosis. This work demonstrates the clinical utility of quantitative statistical techniques in veterinary diagnostics to provide an evidence-based approach for decision making in veterinary histopathology.

## 2. Materials and Methods

### 2.1. Study Population and Data Collection

Records of dogs and cats with bladder histology submitted to The University of Queensland Veterinary Laboratory Service (UQVLS) between January 1994 and March 2016 were obtained via a search enquiry on the histopathology database for canine and feline pathology reports containing at least one of the following terms: bladder, cystitis, transitional cell, urinary, urothelial. The database search process is outlined in Figure 1. Pathology records were excluded if they only mentioned gallbladder (not urinary bladder, *n* = 747), if the record contained cytology but not histology (*n* = 24) or if the animal was less than six months of age (*n* = 46), as immature bladder wall can have umbilical artery or urachal remnants [26]. Records were excluded when urinary bladder had been discussed in the gross section of the necropsy report, but no histology had been taken (*n* = 68), and when no diagnosis could be made due to poor sample size or quality (*n* = 5). Tissue blocks were retrieved (where available, 83 could not be located or the sample was not of diagnostic quality), and slides recut at 4 µm thickness and stained with hematoxylin and eosin (H&E) (Leica ST5020 autostainer).

In addition, bladder pathology records were obtained from the School of Veterinary Medicine, College of Science, Health, Engineering and Education, Murdoch University (MUSVM). The search function on the MUSVM database prevented an identical database search being performed, thus the primary disease category was selected, followed by a search for keywords (cystitis, urin*, transit*, uroth*, urolith*, TCC and UCC) within each year of archived records. Seventy-one blocks were identified, and slides recut and stained with H&E using a manual staining method.

A selection of Brisbane veterinary clinics was recruited for involvement in the project based on their geographical location, and a small number of prospective bladder biopsies were obtained from dogs and cats undergoing cystotomy at those clinics as part of disease diagnosis or treatment (*n* = 19). Finally, eleven slides containing canine or feline urinary bladder specimens were donated by a local veterinary diagnostic pathology company. The UQVLS pathology database search for biopsy and necropsy samples and the prospective collection of biopsy and necropsy samples was performed under the University of Queensland animal ethics ANRFA/SVS/259/16.

### 2.2. Histopathology

A wide range of histologic features were scored for each slide, outlined in Appendix A. Many variables were scored based on presence or absence, such as urothelial inflammation or neoplastic urothelium. Some variables had multilevel scoring, such as primary submucosal inflammatory type (lymphocytic, neutrophilic, lymphoplasmacytic or granulomatous) and submucosal hemorrhage (none, hemorrhage present in up to 25%, 26–50% and >50% of the submucosa) (Appendix A). These histologic features were selected based on a literature review of microscopic changes in urinary bladder diseases in people [27,28,29,30,31]. Retrieved cases were interrogated by EJ (pathology trainee) and RA (Diplomate of the American College of Veterinary Pathologists), and after microscopic slide review, cases were categorized ranked in this order—neoplasia, urolithiasis (bladder stones), cystitis (inflammation in the bladder wall which may be due to infectious or non-infectious causes), normal bladder wall, or other diagnoses.

Based on the human literature [32,33] and a review of 56 normal and diseased bladder specimens at the beginning of this project, we agreed that up to 20 scattered lymphocytes per 100× magnification field of the mucosa and submucosa were considered normal, with greater than 20 leading to classification as cystitis, particularly if there was concurrent hemorrhage and/or edema. Normal bladders could have occasional lymphocytes scattered throughout the urothelium and lamina propria, but no neutrophils.

Urolithiasis is not a histologic diagnosis per se. However, these cases were identified and separated based on clinical imaging and/or cystoscopy, or gross postmortem findings. Records were assigned to the urolithiasis category if uroliths were mentioned in the submitted history regarding the current clinical presentation for biopsy cases, or if they were reported in the necropsy report.

Where neoplasia of any type had been identified in the bladder wall, records were placed into the ‘neoplasia’ category, even if there was concurrent inflammation or urolithiasis, as neoplasia was considered to be the predominant pathologic process. Tissue was categorized as normal if the bladder wall had no abnormalities or had up to 20 scattered submucosal lymphocytes without hemorrhage and edema.

All other diagnoses were grouped together in the ‘other’ category, including hemorrhage only, edema only, autolysis only, peritonitis (inflammation of the bladder serosa), traumatic bladder rupture, and healed scar tissue (Appendix A). Cases with normal bladder tissue and other diagnoses combined were considered as the base category to which the other three categories of diagnostic outcomes were compared in the statistical modeling. The ‘other’ diagnosis group was combined with normal bladders for two reasons. First, to simulate the array of histologic features that may be observed in absence of one of our main three diagnoses; second, to increase the power of this group as the normal bladder cases alone were a relatively small group. Data was managed using Microsoft Excel (Microsoft Corporation, Redmond, WA, USA).

### 2.3. Statistical Analysis

Multinomial logistic regression analysis was utilized to analyze univariable associations between risk factor variables—histologic variables and species; and the dependent variable, the diagnoses—cystitis, neoplasia or urolithiasis, compared to normal and other diagnoses combined. Stata 15.0 (StataCorp., College Station, TX, USA) was used for all statistical analyses. The urothelial inflammation category was changed from a multilevel grading to the dichotomous ‘presence or absence’ of urothelial inflammation for the multivariate model, due to low case numbers in the higher categories. Likewise, for submucosal hemorrhage—categories 3 (26–50%) and 4 (>50% submucosa containing hemorrhage) were combined due to a low number of cases in category 4. A multiple Wald test was used to evaluate the statistical significance of all categories together for any histologic variable [10]. Variables for which Wald’s *p* < 0.10 were considered for multivariate analysis.

For the multivariable model, forward stepwise model building procedure was performed and the overall *p* values and odds ratios (OR) with 95% confidence intervals (CI) for explanatory variables were recorded. The stepwise selection process was stopped once all covariates were significantly (*p* ≤ 0.05) contributing to the model in any of the four outcome categories (cystitis, neoplasia, urolithiasis, baseline of normal/other). First order interactions between explanatory variables were also explored. Variables not selected for the original multivariate model were added back one at a time, with significant variables (*p* ≤ 0.05) retained. Using this approach, we were able to identify variables that by themselves were not significantly related to the outcome, but made an important significant contribution in the presence of other variables [34]. The Hausman–McFadden Test was used to test the assumption that the model odds ratios for each level of the histologic variable were independent of the other levels [35].

Several approaches were used to conduct regression diagnostics. First, three binary logistic regression models were created from the final multivariate model (baseline/cystitis, baseline/neoplasia, baseline/urolithiasis) and regression diagnostics were carried out on each model as described [10]. Thus, covariate patterns with outlying standardized Pearson residual and delta–beta values were identified, and the models were then rerun excluding cases from within these patterns and the changes in the resulting coefficients were examined. Model fit was then assessed using the Hosmer–Lemeshow goodness-of-fit test. Secondly, a recently developed overall goodness-of-fit-test for multinomial logistic regression models [36] was applied using the—logitgof—command in Stata (StataCorp., College Station, TX, USA). Predicted probabilities were calculated from the multinomial model using the—margins—command in Stata, where the predicted probability of each diagnostic outcome was calculated at each level of the individual variable and variable combinations, while all other variables in the model were at their means.

## 3. Results

The dataset consisted of 338 cases (Table 1) including 102 cystitis (30% of all samples), 84 neoplasia (25%, 71% of these UC, Appendix A), 42 urolithiasis (12%), 63 normal bladders (19%) and 47 other diagnoses (14%). During the slide review process, 23 cases had their diagnosis changed. The most common reason for a changed diagnosis was reclassifying a cystitis diagnosis as normal due to paucity of inflammatory infiltrates (*n* = 7).

Stepwise construction of the multinomial logistic regression model revealed six variables that were significantly associated with disease diagnosis, compared to the normal/other disease baseline category (herein referred to as ‘baseline’), when other variables were accounted for: species, urothelial ulceration, urothelial inflammation, submucosal lymphoid aggregates, neutrophilic submucosal inflammation, and having a moderate amount of submucosal hemorrhage (Table 2).

Species was significant for the urolithiasis outcome, with dogs having at least five times higher odds of having urolithiasis than cats (*p* = 0.04). Slides with urothelial ulceration were 13 times more likely to be diagnosed with urolithiasis (*p* < 0.01) compared to slides without urothelial ulceration. The presence of inflammatory cells infiltrating the urothelial cell layer was significantly associated with all disease groups, particularly urolithiasis (*p* < 0.01). Urothelial inflammation was also associated with neoplasia (*p* < 0.01) and cystitis (*p* = 0.02) compared to cases that did not have urothelial inflammation (Table 2).

The presence of submucosal lymphoid aggregates was significantly associated with cystitis (*p* = 0.01) and neoplasia (*p* = 0.03). Neutrophilic submucosal inflammation was significantly associated with all disease groups compared to the baseline category, particularly urolithiasis where slides with neutrophilic submucosal inflammation were 17 times more likely to belong to the urolithiasis group (*p* = 0.01). Neutrophilic submucosal inflammation was also strongly associated with cystitis (*p* < 0.01), and moderately associated with the neoplasia category (*p* = 0.02). A ‘moderate to severe’ amount of hemorrhage (>26% of the submucosa contains extravasated erythrocytes) was significantly associated with cystitis (*p* = 0.01). Predicted probabilities for the significant variables (*p* ≤ 0.05) are displayed in Appendix A and are shown graphically in Figure 2. Each data entry (vertical line) depicts a disease diagnosis (cystitis, neoplasia, urolithiasis and normal/other) without and with the significant histologic variable.

## 4. Discussion

Five histologic variables and one animal variable were found to be significantly associated with one or more disease group using logistic regression modeling—urothelial inflammation, urothelial ulceration, type of submucosal inflammation, amount of submucosal hemorrhage, presence of submucosal lymphoid aggregates and species. Our result of dogs having five times higher odds of being diagnosed with urolithiasis compared to cats is not surprising as urolithiasis is reported more commonly in dogs than cats [37]. Urolithiasis is thought to be present in up to 1.5% of cats seen at veterinary practices and up to 3% of dogs [15], which is reflected in our findings.

Urothelial ulceration was significantly associated with urolithiasis, which makes sense pathophysiologically as this disease is characterized by the presence of a urolith in the bladder lumen. The bladder stones cause physical trauma to urothelial cells, sometimes to the point of complete urothelial loss [15,38]. It is important to note that urothelial loss may also occur as a postmortem change or as part of tissue handling during biopsy. However, we endeavored to prevent overlap here by using separate variables for urothelial loss (denudation) and ulceration. Postmortem or tissue handling-induced urothelial loss showed minimal tissue reaction, suggesting that it was artefactual, as opposed to pathologic ulceration which includes one or more of the following cellular changes in addition to the urothelial loss—urothelial cell flattening and sliding, submucosal or urothelial inflammation, submucosal hemorrhage or submucosal edema in the remaining urothelial layer and/or submucosa [39].

The presence of urothelial inflammation was strongly associated with all disease groups but could not be used to differentiate between them. It is normal to have low numbers of resident lymphocytes within the urothelium [33]. However, it makes biological sense that numbers would increase in response to physical trauma such as uroliths, the presence of microorganisms, or in response to ‘foreign’ neoplastic cells (most neoplasms in this dataset were urothelial carcinomas which arise from urothelial cells). In addition, neoplasms weaken tissue architecture and can make the tissue more sensitive to trauma or infection, further increasing leukocyte numbers. Finally, neoplasms are known to recruit specific intra-tumoral lymphocytes, which may be what we observed in our UC samples [40].

Neutrophilic submucosal inflammation was increased in association with all disease outcomes while non-neutrophilic inflammation (mononuclear; primarily lymphocytic or lymphoplasmacytic infiltrates) was not significantly associated with any disease group compared to baseline. As mentioned, the presence of scattered lymphocytes in the urothelium and submucosa can be normal [32,33]. However, neutrophils are not part of the resident leukocyte population and are only present when stimulated by inflammation, which is consistent with our results. Therefore, the submucosa of a normal bladder wall should not contain any neutrophils.

The most severe categories of hemorrhage observed in the histologic analysis (a moderate to marked amount of hemorrhage—extravasated erythrocytes occupying >26% of the submucosal area) were significantly associated with the cystitis outcome when compared to baseline. Submucosal hemorrhage was observed in all disease groups, but it was highest in the cystitis group. This is likely attributable to the vasoactive cytokines released during acute inflammation, causing increased vascular permeability and leakage [41]. We observed a high level of neutrophilic inflammation in the cystitis group; submucosal neutrophils are likely to be associated with hemorrhage because one of the enzymes released, elastase, can contribute to the breakdown of arteriolar walls [42]. In addition, bacteria can induce endothelial damage, which may result in submucosal hemorrhage in some bacterial cystitis cases [39]. Submucosal hemorrhage may occur as a sequela to tissue handling during biopsy. However, tissue handling does not result in extravascular inflammatory cell infiltrates.

Finally, the presence of submucosal lymphoid aggregates was associated with cystitis and neoplasia, but not with urolithiasis. Aggregates of lymphocytes (often with germinal centers) are a part of normal immune surveillance in many tissues [43,44], but in the human urinary bladder are thought to represent chronic inflammation [45]. There are no published standards for the role of lymphoid aggregates in canine or feline bladders. However, the chronic inflammation theory is plausible for these species as well. Chronic inflammation is frequent in cases of recurrent cystitis [46,47], while tumor-infiltrating leukocytes are commonly observed alongside neoplasia [48]. Animals with recurrent bacterial cystitis may have predisposing breed, anatomical or metabolic factors that have caused them to have bladder inflammation more frequently throughout their life prior to histologic sampling [46,47,49,50], thus resulting in the presence of lymphoid aggregates. Uroliths may take a more acute clinical course, with animals showing clinical signs more acutely than those with low-grade cystitis or with neoplasia; this could theoretically lead to earlier detection of disease and therefore tissue samples being obtained earlier in the disease process before lymphoid aggregates developed. Lastly, there is likely to be little antigenic drive in urolithiasis to provoke an acquired immune response, which typically occurs in lymphoid tissue. Conversely, infection and neoplasia are associated with abnormal self or non-self-antigens, and therefore would be more likely to trigger an acquired, lymphocytic immune response [51,52] and thus develop lymphoid aggregates.

When bladder stones were present, the histologic diagnosis made by the initial pathologist was often ‘cystitis’, because bladder wall inflammation frequently occurs in these cases. Urolithiasis is not a histologic diagnosis. However we chose to separate these records into a different group for the modeling process, ‘urolithiasis,’ and not include them in the ‘cystitis’ category as the inflammation can be attributed to the physical trauma of the stone in addition to potential bacterial infection. It was recognized that bacterial cystitis may lead to urolithiasis, namely struvite stone formation secondary to urease-producing bacteria [53,54]. Conversely, uroliths can impair bladder defenses, facilitating the development of bacterial cystitis [53]. Up to 50% of canine bladder stones are thought to be infection related [55], a phenomenon that is much more common in dogs than in cats [56]. The small number of cases in this study with confirmed concurrent bacterial cystitis and urolithiasis were categorized with other urolithiasis cases instead of other bacterial cystitis cases for several reasons. Firstly, urine culture for bacterial detection may not be performed routinely on every urolith case, and therefore the urolithiasis category in this dataset is likely to already contain some cases with both urolithiasis and bacterial urinary tract infection. Secondly, the cystitis disease group is already a heterogenous combination of (a) cases with bacterial or fungal infection confirmed by urine culture, (b) cases with suspected bacterial infection without bacterial culture, and (c) cases of suspected feline idiopathic cystitis (no infectious organism involved). Thirdly, we hypothesized that the mechanical damage caused by uroliths may lead to a different inflammation profile to bacterial cystitis alone, and therefore cases with uroliths and bacterial infection were deemed more similar to the urolithiasis cases than cases of cystitis without bladder stones. The authors realize that there is likely to be overlap between the cystitis and urolithiasis groups in this study and it would be ideal to have urine culture results for each of these cases to enable more accurate stratification of these diseases. However, this was not possible due to the retrospective nature of this study.

One limitation of this work is that cases of chronic cystitis are more likely to undergo clinical biopsy than acute cystitis patients, so there is likely to be some skewing of our dataset to contain a higher proportion of chronic cystitis cases than would occur in the general pet population. For the cystitis outcome, it is important to note that our definition of cystitis (histologic evidence of inflammation in the bladder wall) is different from the commonly used clinical definition of cystitis (patients with lower urinary tract clinical signs and often with positive bacterial culture indicating urinary tract infection). Due to the histologic and primarily retrospective nature of this study, many cases were lacking urine culture results, making it impossible to definitively categorize those cases as having or not having bacterial cystitis. Further, the combination of normal and other diagnoses is likely to have had some impact on the results. The two groups were combined to increase case numbers in the baseline category and to replicate the range of other diseases that may be compared to our disease groups. In future it would be prudent to recruit a larger sample size to enable the comparison of disease categories to a group of normal bladders without any other disease combined with it. The small sample size for cats in our study is thought to be a sampling bias as a result of a combination of factors; fewer cats undergoing postmortem examination at our institution, the difficulty of obtaining bladder biopsies from cats, particularly males [57], as well as a lower proportion of owned cats compared to dogs in Australia; 27% of households have a pet cat, 40% of households have a pet dog [58].

The primary limitation of this study is its retrospective nature. A retrospective study design was chosen for two reasons. Firstly, despite bladder disease occurring commonly in dogs and cats, bladder biopsy is not frequently performed and therefore a suitable sample size of bladder histologic specimens would be very difficult to attain via a prospective study. This particularly applies to the rarely biopsied but highly clinically significant syndrome of feline idiopathic cystitis. Secondly, evaluating 20 years of archived pathology records provides a detailed insight into the historical nature of bladder disease samples—an analysis that has not previously been performed. A limitation of this approach is that ideally, mathematical models should be developed from prospective data to minimize missing data and tighten the definition of subject groups [18]. However, financial and ethical constraints prohibited a prospective study in this instance. Some limitations for using logistic regression modeling are that odds ratios are the only output, the sequence of entering variables into the model is based purely on the statistical criteria, and it can be difficult to interpret interactions between variables [59]. In addition, the use of logistic regression means that our probabilities for disease outcome based on histologic variables were calculated by keeping all other variables in the model at their mean value.

## 5. Conclusions

In summary, we applied logistic regression modeling to a histology dataset of canine and feline urinary bladders from Eastern and Western Australia which identified six significant variables that were associated with disease outcome compared to baseline; urothelial ulceration, urothelial inflammation, neutrophilic submucosal inflammation, submucosal lymphoid aggregates, amount of submucosal hemorrhage, and species. The next step in this work will be the creation of a predictive probability tool for bladder disease diagnosis and validation of the tool in a prospective controlled setting, to evaluate its potential use for diagnostic veterinary pathology applications. Going forward, a prospective study evaluating both histologic features of canine and feline urinary bladder disease, and patient outcome information would be required to refine the accuracy and clinical applicability of this model and predictive tool.

## Figures and Tables

**Figure 1 vetsci-07-00190-f001:**
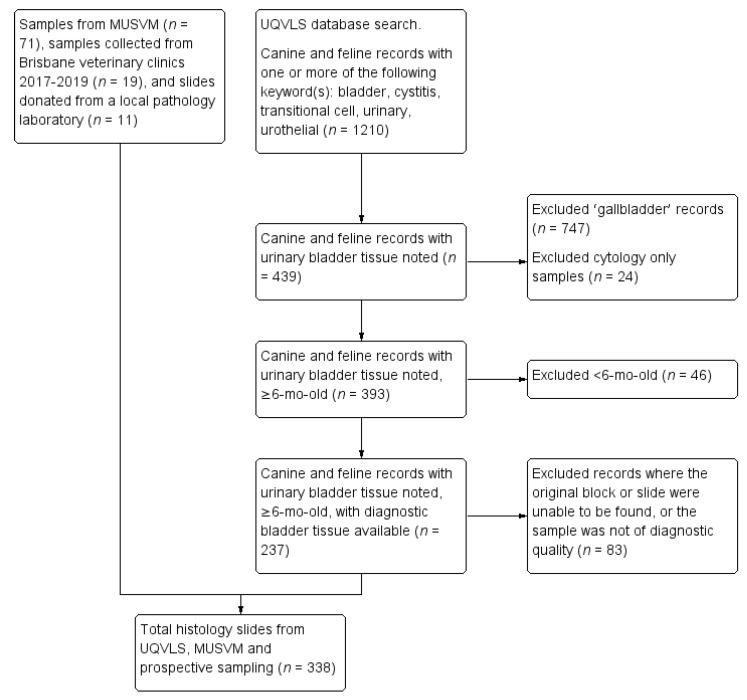
Flow chart for the selection of canine and feline urinary bladder histology slides submitted to the University of Queensland Veterinary Laboratory Service (UQVLS) between January 1994 and March 2016, selected slides from the School of Veterinary Medicine College of Science, Health, Engineering and Education (MUSVM), Murdoch University pathology archives, and prospective samples obtained from local veterinary clinics and a veterinary pathology company in the Brisbane region.

**Figure 2 vetsci-07-00190-f002:**
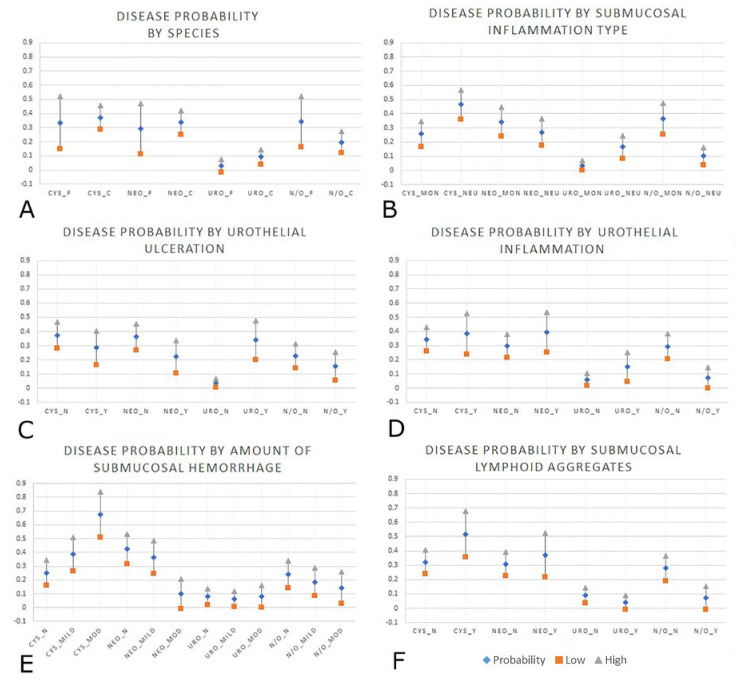
Graphical representations of the six significant variables from the logistic regression modeling, all with 95% CI, displaying probability (diamond) with the low (square) and high (triangle) ends of the CI. (**A**): Probability for bladder diagnosis by species; cystitis (CYS), neoplasia (NEO), urolithiasis (URO), normal/other (N/O), feline (_F), canine (_C). (**B**): Probability for bladder diagnosis by type of submucosal inflammation; primarily mononuclear (non-neutrophilic) inflammation (_MON), primarily neutrophilic inflammation (_NEU). (**C**): Probability for bladder diagnosis by presence or absence of urothelial ulceration; No urothelial ulceration (_N), yes, urothelial ulceration present (_Y). (**D**): Probability for bladder diagnosis by presence of urothelial inflammatory infiltrate; No urothelial inflammation (_N), Urothelial inflammation was present (_Y). (**E**): Probability for bladder diagnosis by amount of submucosal hemorrhage; No submucosal hemorrhage (_N), Mild amount of submucosal hemorrhage (_mild), Moderate to severe amount of submucosal hemorrhage (_mod). (**F**): Probability for bladder diagnosis by presence or absence of submucosal lymphoid aggregates; No submucosal lymphoid aggregates (_N), Yes, submucosal lymphoid aggregates were present (_Y).

**Table 1 vetsci-07-00190-t001:** Count data of all reviewed histology slides by species, following slide review.

Diagnosis	Canine (%)	Feline (%)	Total (%)
Cystitis	81 (30)	21 (30)	102 (30)
Neoplasia	74 (28)	10 (14)	84 (25)
Other	34 (13)	13 (18)	47 (14)
Normal	40 (15)	23 (32)	63 (19)
Urolithiasis	38 (14)	4 (6)	42 (12)
	267	71	338

**Table 2 vetsci-07-00190-t002:** Results of the final multivariate multinomial logistic regression model showing animal or histologic factors associated with diagnoses (cystitis, neoplasia and urolithiasis) compared with the reference category of normal/other diagnoses combined.

	Diagnosis					
Variable	Cystitis		Neoplasia		Urolithiasis	
	OR (95%CI)	*p*	OR (95%CI)	*p*	OR (95%CI)	*p*
Species						
Feline	Reference		Reference		Reference	
Canine	1.95 (0.71, 5.31)	0.19	2.03 (0.75, 5.52)	0.17	5.41 (1.07, 27.46)	0.04
Urothelial ulceration						
No	Reference		Reference		Reference	
Yes	1.12 (0.43, 2.89)	0.82	0.90 (0.33, 2.43)	0.83	13.45 (3.73, 48.56)	<0.01
Lymphoid aggregates						
No	Reference		Reference		Reference	
Yes	6.16 (1.59, 23.87)	0.01	4.59 (1.20, 17.61)	0.03	1.71 (0.32, 9.12)	0.53
Neutrophilic submucosal inflammation						
No	Reference		Reference		Reference	
Yes	6.47 (2.81, 14.88)	<0.01	2.82 (1.23, 6.49)	0.02	17.05 (5.26, 55.23)	0.01
Amount of submucosal hemorrhage						
1	Reference		Reference		Reference	
2	1.98 (0.81, 4.82)	0.13	1.12 (0.48, 2.57)	0.80	1.01 (0.3, 3.39)	0.99
3	4.46 (1.43, 13.93)	0.01	0.39 (0.09, 1.76)	0.22	1.7 (0.40, 7.30)	0.47
Urothelial inflammation						
No	Reference		Reference		Reference	
Yes	4.56 (1.35, 15.42)	0.02	5.45 (1.7, 17.52)	<0.01	10.23 (2.49, 42.06)	<0.01

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
