# Peer review of "Predicting Diagnosis of Australian Canine and Feline Urinary Bladder Disease Based on Histologic Features"

_vetsci, 2020, doi:10.3390/vetsci7040190_

Round 1

Reviewer 1 Report

  1. This is a well written, organized, and conceived study. Figure 1 is useful and a welcome change from other forms that might show flow of choices. I am not an expert in statistics, and so my comments are limited to choices of samples and histologic classification scheme. I find the correlation of disease with type of submucosal inflammation particularly helpful, and I appreciated the usefulness of the data. I am curious about whether this would be valid regardless of the type of biopsy (method) or necropsy sample, and whether or not the location of the biopsy could or would matter for, in particular, inflammatory reaction or lymphoid response. The authors did an excellent job of pointing out study limitations.
  2.  
  3. lines 63-65- The following statement: "Bladder diseases account for 7% of cats hospitalized in veterinary clinics in the United States. An estimated 14% of North American dogs are affected by a urinary tract infection at some point during their lifetime" should be correlated with the studyt set. ie that 267 vs. 71 canine vs. feline samples were analysed.Is it more likely that canines are sampled vs. felines? What percentage of hospitalized canine cases are urinary (to compare directly to the cat statement).

  4. line 188: "The dataset consisted of 338 cases (Error! Reference source not found.)" ? This also repeated on line 200, 207
  5.  

Author Response

Reviewer 1

This is a well written, organized, and conceived study. Figure 1 is useful and a welcome change from other forms that might show flow of choices. I am not an expert in statistics, and so my comments are limited to choices of samples and histologic classification scheme. I find the correlation of disease with type of submucosal inflammation particularly helpful, and I appreciated the usefulness of the data. I am curious about whether this would be valid regardless of the type of biopsy (method) or necropsy sample, and whether or not the location of the biopsy could or would matter for, in particular, inflammatory reaction or lymphoid response. The authors did an excellent job of pointing out study limitations.

Thank you for your time and feedback. Regarding the type of sampling (biopsy or necropsy), our modelling process found that this was not a significant variable, therefore according to the model our findings would have been the same regardless of the sampling type. To address your comment on the location of the biopsy site - I have been unable to find anything in the literature to suggest that this makes a difference. Having used mostly retrospective data we do not have access to the site of biopsy for our samples, however I agree this would make an interesting component of a prospective study.

lines 63-65- The following statement: "Bladder diseases account for 7% of cats hospitalized in veterinary clinics in the United States. An estimated 14% of North American dogs are affected by a urinary tract infection at some point during their lifetime" should be correlated with the studyt set. ie that 267 vs. 71 canine vs. feline samples were analysed. Is it more likely that canines are sampled vs. felines?

We hypothesised due to our sample numbers that canines are more likely to be sampled than felines, likely based on the difficulty of obtaining bladder biopsies from cats and also the percentage of dog vs cat ownership which I have added to the manuscript on lines 348-352.

What percentage of hospitalized canine cases are urinary (to compare directly to the cat statement). Thank you for pointing this out - I had been unable to find a direct comparison. There is a paper on primary accession cases in the UK that found 3.2% of canine presentations to be attributable to the urinary tract, however this includes renal disease as well (reference below). I had not included a comparison due the apparent absence of one in the literature.

O Neill, D. G., Church, D. B., McGreevy, P. D., Thomson, P. C., & Brodbelt, D. C. (2014). Prevalence of disorders recorded in dogs attending primary-care veterinary practices in England. PloS one, 9(3), e90501. https://doi.org/10.1371/journal.pone.0090501

line 188: "The dataset consisted of 338 cases (Error! Reference source not found.)" ? This also repeated on line 200, 207 Thank you, this error has been resolved.

Reviewer 2 Report

These authors have provided an interesting well-described retrospective assessment of the histological features of canine and feline urinary bladder disease. 

Specific Comments   The authors appropriately discuss the limitations of their study based upon its retrospective versus prospective design

The text has odd inserts in 3 places---"Error! ----   " with regard to Supplemental Table 2 (line 145), data set, line 188, and the statement on line 207.  Please clarify.

Reference # 25 has no year listed. Please complete.

Author Response

Specific Comments The authors appropriately discuss the limitations of their study based upon its retrospective versus prospective design The text has odd inserts in 3 places---"Error! ---- " with regard to Supplemental Table 2 (line 145), data set, line 188, and the statement on line 207. Please clarify.

Thank you, this error has been resolved.

Reference # 25 has no year listed. Please complete.

This has been resolved, thank you.